# TEXTBOOK CONSISTENCY WEIGHTED INTERNET IM-PROVES EFFICIENCY TWOFOLD

## ABSTRACT

We propose a novel method, Textbook Consistency, to improve the training efficiency of large language models by leveraging textbooks as a guiding signal for learning from internet-scale data. Rather than relying on hard filtering of data based on quality thresholds before training, our approach adaptively adjusts the weight of data during training based on its consistency with textbooks during training. We compute the cosine similarity between internet data and textbooks in a latent space, using this metric to modulate the cross-entropy loss. Our method significantly enhances training efficiency, achieving twice the effectiveness by reducing training time or the number of tokens required. Empirical results show superior performance on language models trained on large datasets like FineWeb and The Pile, with extensions to other domains such as robotics. Our method is simple to implement, incurs no additional overhead, and is compatible with existing data curation techniques.

## 1 INTRODUCTION

The internet provides a vast and diverse pool of knowledge, making it a critical resource for advancing large language models (Brown et al., 2020; Hoffmann et al., 2022; Kaplan et al., 2020). Training these models on internet-scale data takes months and requires thousands, if not tens of thousands, of GPUs (Touvron et al., 2023a; Dubey et al., 2024). To make training more efficient and improve data quality, researchers have focused on methods to data filtering, duplicate removal, perplexity-based filtering, hand curation, identifying new data sources (see e.g. Lee et al., 2021; Penedo et al., 2023; Computer, 2023; Li et al., 2024; Soldaini et al., 2024; Gao et al., 2020; Soboleva et al., 2023; Albalak et al., 2024a). A dominant strategy involves filtering data by comparing it to smaller, high-quality sources such as textbooks, and filter out or keep data in hard manner (Brown et al., 2020; Wenzek et al., 2019). This strategy is frequently applied in the training of state-of-the-art language models such as Llama (Touvron et al., 2023a) and datasets such as RefinedWeb (Penedo et al., 2023). However, this process requires training proxy LLMs on data filtered at different thresholds, which is a less scalable and tedious process. Moreover, this method imposes a hard threshold, meaning data is either fully retained or discarded before training, without allowing for a more nuanced approach to learning or unlearning based on data's relevance and importance.

In this paper, we explore whether training efficiency can be improved by adaptively weighting internet data during training, based on its consistency with textbook-quality sources. We propose to use textbooks to guide learning from internet during training. The intuition is that internet provide diverse learning signals while textbooks provide high quality guidance – the model should learn more if the data is consistent with target data or vice versa. We measure the consistency between internet (source data) and textbooks (target data) by computing the cosine similarity between them in a latent space, and weighting next token prediction cross entropy loss with cosine similarity. Since our method aims to upweight and downweight data adaptively based on their consistency with textbook, we denote our method as Textbook Consistency.

Empirically, we found that our method performs well in language model training at no measurable additional cost. When using textbooks, which consist of high-quality public instruction tuning datasets, as a consistency target for learning from large and diverse internet-scale datasets like FineWeb (Penedo et al., 2023) and The Pile (Gao et al., 2020), our approach achieves significantly lower validation loss, better performance in downstream tasks, and a superior scaling trend. Our

**Figure 1** **Overview of Textbook Consistency method for training language models**. The method learns from internet by comparing them to high-quality textbook sources. The loss function is weighted based on the consistency between internet and textbook data. This approach adaptively upweights or downweights data to improve training efficiency and model performance without adding significant computational cost.

method enables twice the training efficiency, meaning it can reduce training time or, equivalently, the number of training tokens by half. This demonstrates the effectiveness of adaptively upweighting and downweighting data based on their consistency with target datasets. We applied our method to tasks beyond language modeling, such as learning robotics behaviors from large unsupervised exploration data (Yarats et al., 2022), where it learns from large and diverse unsupervised exploration trajectories, with consistency provided by a small but highly rewarding demonstration. Our method demonstrates similar advantages, outperforming prior methods by a large margin and achieving higher reward.

Our empirical evaluations demonstrate the effectiveness of Textbook Consistency, highlighted below.

- We show Textbook Consistency can leverage high quality textbook to adaptively weight internet data for efficient training with up to 2-3x efficiency.
- We show that Textbook Consistency performs well across various manually curated internet datasets, demonstrating that our method is compatible with existing data curation approaches.
- We show that our method can enable higher accuracy in downstream language tasks and show application in non-language domains.

## 2 METHOD

We introduce a method called Textbook Consistency that enhances learning from large-scale and diverse internet data by aligning it with narrow but high-quality textbook datasets. The core idea behind this approach is to guide the learning process using the structure and accuracy of textbooks while drawing from the vastness and diversity of internet sources. To achieve this, we use two primary data sources: large, general-purpose internet datasets, which provide a broad range of information, and curated textbook datasets, which are more focused and domain-specific but contain high-quality content.

We consider next token prediction objective which is to predict the probability distribution of the next word in a sequence given the preceding words. Let's denote the training data as a sequence of tokens $\{x_1, x_2, \ldots, x_T\}$. For each token $x_t$, the model aims to maximize the conditional likelihood of token $x_{t+1}$ given all the previous tokens, $\{x_1, \ldots, x_t\}$. The objective function is the negative log-likelihood (NLL) over all tokens in the dataset. Given a sequence of tokens $\{x_1, x_2, \ldots, x_T\}$, the training loss is defined as:

$$\mathcal{L}_{\text{NLL}} = -\sum_{t=1}^{T} \log P(x_{t+1}|x_1, \ldots, x_t; \theta)$$

where $P(x_{t+1}|x_1, \ldots, x_t; \theta)$ is the probability distribution over the vocabulary, predicted by the model's parameters $\theta$.

Incorporating Textbook Consistency into the next token prediction framework introduces a weighted loss mechanism to improve the learning from diverse internet data by considering the similarity between internet-sourced sentences and those from high-quality textbooks. Let's denote two datasets: *internet*: $\mathcal{D}_{\text{internet}} = \{s_i^{\text{internet}}\}$, where each $s_i^{\text{internet}}$ represents a sentence sampled from the internet. *textbook*: $\mathcal{D}_{\text{textbook}} = \{s_j^{\text{textbook}}\}$, where each $s_j^{\text{textbook}}$ represents a sentence sampled from a high-quality textbook.

In the weighted next token prediction setup, a random mini-batch of sentences from the internet, denoted as $\{s_1^{\text{internet}}, s_2^{\text{internet}}, \ldots, s_N^{\text{internet}}\}$, is compared to a random mini-batch from textbooks, denoted as $\{s_1^{\text{textbook}}, s_2^{\text{textbook}}, \ldots, s_M^{\text{textbook}}\}$, using cosine similarity.

The cosine similarity between two sentence embeddings $e(s_i^{\text{internet}})$ and $e(s_j^{\text{textbook}})$, where $e(\cdot)$ represents the sentence embeddings produced by an embedding model, is computed as:

$$\text{cosine}(e(s_i^{\text{internet}}), e(s_j^{\text{textbook}})) = \frac{e(s_i^{\text{internet}}) \cdot e(s_j^{\text{textbook}})}{\|e(s_i^{\text{internet}})\| \|e(s_j^{\text{textbook}})\|}.$$

Each sentence from the internet dataset is weighted by its average cosine similarity with the sentences in the textbook mini-batch, based on their embeddings. Let the weight for the $i$-th internet sentence be denoted as $w_i$, where:

$$w_i = \frac{1}{M} \sum_{j=1}^{M} \text{cosine}(e(s_i^{\text{internet}}), e(s_j^{\text{textbook}})),$$

where the embedding $e$ comes from a pretrained embedding model such as BERT, or from the model itself. Incorporating these weights into the next token prediction task, the loss for internet-sourced data becomes a weighted negative log-likelihood:

$$\mathcal{L}_{\text{weighted}} = - \sum_{i=1}^{N} w_i \sum_{t=1}^{T_i} \log P(x_{t+1}|x_1, \ldots, x_t; \theta).$$

This approach ensures that sentences from the internet, are weighted according to their cosine similarity with sentences from textbooks. This ensures that the model can learn effectively from large-scale, diverse internet data while being guided by high-quality textbooks.

The method is illustrated in Figure 1, and the corresponding algorithm is shown in Algorithm 1.

---

**Algorithm 1** Learning On Internet With Textbook Consistency

---

**Required:** Internet dataset $D$, Textbook dataset $T$, Model $M$, Embedding Model $E$.
Initialize
**for** Training Iterations **do**
    Sample a mini batch from internet dataset $D$
    Sample a mini batch from textbook dataset $T$
    Compute embeddings for both batches with $E$
    Compute average cosine similarity between $D$ and $T$ in embedding space
    Update model $M$ to minimize weighted cross-entropy loss
    (Optional) Update embedding $E$
**end for**
Final model

---

## 3 EXPERIMENT

Our study is based on the LLaMA (Dubey et al., 2024) architecture, and we consider model sizes of 375M, 1.2B, and 3B in our experiments. Although we explore larger models like the 3B size, the majority of our experiments focus on the 1.2B model. The implementation is in Jax/Flax (Bradbury et al., 2018; Heek et al., 2023). We use batch sizes of 0.5M and 1M, sampling a batch size of 0.5M from a narrow domain. For embedding both the source and target, we utilize BERT-base (Devlin et al., 2018) from sentence-transformers (Reimers and Gurevych, 2019). We swap learning rate

with grid search. Unlike standard language model training, where sentences are packed together to maximize FLOP utilization on GPUs/TPUs, our approach requires computing sentence embeddings and using embedding similarity to weight the next-token prediction loss. To prevent cross-sentence attention during next-token prediction, we compute the embeddings after loading the data. We then pack the sentences and apply an attention segmentation mask to ensure that attention is restricted within each sentence, with no interaction between different sentences.

We use muP (Yang et al., 2022) to parameterize the model and conduct proxy experiments to confirm that the learning rate and other hyperparameters optimized for our small 375M model can be successfully applied to larger models. When applying our hyperparameter search method to the C4 training set, our 375M baseline model achieves a test set loss of 2.58 on C4, which improves upon the 2.7 loss reported in Appendix G of Chinchilla (Hoffmann et al., 2022) for a similarly sized model. This result demonstrates the strength of our baseline model compared to the current state of the art.

We report the number of tokens consumed during training. The computational cost (FLOPs) incurred by the embedding model is less than 0.5% of the total training FLOPs, even for the smallest 375M model. This percentage decreases further for larger models, such as the 1.2B, 3B, and other larger LLMs. For all methods, our experiments are conducted on 64 TPUv4 chips on Google Cloud, equivalent to 32 Nvidia A100. We use bf16 for activation and fp32 for parameters and gradients. We use the AdamW optimizer (Loshchilov, 2017) with max gradient norm 1.0. For the learning rate schedule, we use linear warmup and cosine decay.

We apply Textbook Consistency to two datasets: FineWeb and The Pile. The Pile is an earlier, widely-used dataset in the community. FineWeb, on the other hand, is a recent state-of-the-art, high-quality dataset, carefully curated through both manual and model-based filtering techniques.

We list some details about the high-quality datasets used as textbooks.

- *OpenHemes* is a high-quality dataset (Teknium, 2023). The dataset consists of questions and answers sourced from benchmarks and user and AI model conversations. Each turn in a conversation has two fields: a "from" field, which denotes the role of that turn, and a "value" field, which contains the actual text. For our embedding purposes, we format each conversation as a sentence.
- *MetaMathQA* is a math-focused dataset (Yu et al., 2023) that contains math questions and answers.
- *EvoInstruct* is a dataset containing conversations on various topics (Xu et al., 2023). Similar to the above, we format each conversation as a sentence before computing embeddings.

Other high-quality textbook datasets, such as OpenOrca (Lian et al., 2023) and UltraChat (Ding et al., 2023), can be also included into textbook, but we leave them for future work. We combine *EvoInstruct*, *MetaMathQA*, and *OpenHemes*. We found that it is also important to mix in high-quality internet datasets, such as *C4* (Raffel et al., 2020), to achieve higher diversity. The corresponding ablation study is presented in the experimental section. We randomly sampled 150M tokens from *C4*, combined with about 50M tokens from the pre-processed textbooks, giving a total of 200M tokens in the post-processed dataset, which we use for our *Textbook Consistency* training.

### 3.1 EVALUATION RESULTS

The evaluation section is divided into performance with different data sizes, performance with different model sizes, downstream evaluation, and ablation evaluation.

#### 3.1.1 TRAINING EFFICIENCY

The graph in Figure 2 shows the effect of different training data and methods on validation loss, using a textbook holdout set for evaluation. Three distinct curves represent different configurations, with each data point corresponding to a full training run, *i.e.*, 10B token point and 20B token point each denote training on 10B and 20B tokens, respectively.

- **Textbook data**. The blue curve, representing the baseline, shows training on multiple epochs of textbook data (i.e., repeated textbook data). Initially, validation loss decreases as the number of tokens increases, but this trend reverses as the model begins to overfit due to excessive repetition. This demonstrates that although textbook data is high-quality, its limited quantity makes it ineffective for training a well-performing model on its own.

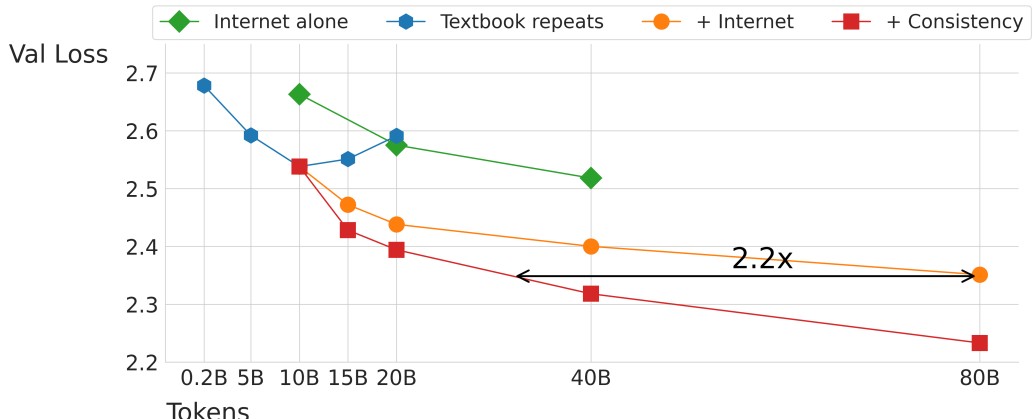

**Figure 2   Validation loss as a function of data size**. Each data point in this figure is a full training run, and evaluated on a hold out subset of textbook. The blue curve repeating 200M textbook data which is a mixture of C4 subset and high quality instruction tuning datasets for multi epoch training. The orange curve denotes adding internet data (RefineWeb) to enlarge training set. The red curve denotes our training method by incorporating the adaptive consistency between textbook and internet.

- **Internet data**. The orange curve, which incorporates internet data (RefineWeb), specifically shows that the starting point of its overlap with the blue curve represents zero internet data, relying solely on repeated textbook data. As more internet data is added, the validation loss decreases compared to using textbook data alone, highlighting the benefit of a larger and more diverse training set.

- **Textbook Consistency**. The red curve represents the model using adaptive consistency between textbook and internet data, and it achieves the lowest validation loss, with the most significant improvement as the number of tokens increases. Notably, at the point where the model has processed 40B tokens, the red curve outperforms the others by a factor of 2x in terms of validation loss reduction, indicating the effectiveness of incorporating consistency between datasets. This suggests that balancing and refining the dataset with consistency techniques can lead to better model performance on unseen textbook data.

> **Takeaway:** Our method surpasses the state-of-the-art, achieving more than double the training efficiency, with the performance gap widening as scale increases.

### 3.1.2   GENERALIZATION ACROSS MODEL SIZE

Figure 3 presents the results of evaluating models of different sizes (375M, 1.2B, and 3B parameters) on validation loss, highlighting the impact of incorporating additional data and consistency techniques. The blue curve represents models trained solely on textbook data, showing a gradual decrease in validation loss as the model size increases, but consistently yielding the highest validation loss compared to other methods. The orange curve, which includes additional internet data, achieves lower validation losses than textbook-only models, indicating the benefits of using a more diverse dataset for training. The red curve, which incorporates textbook and internet data along with an adaptive consistency method, consistently outperforms the other two approaches, demonstrating the lowest validation loss across all model sizes. Notably, the margin of improvement provided by the consistency approach becomes more pronounced as model size increases, with the 3B model showing the most significant reduction in validation loss. This suggests that the adaptive consistency method scales effectively with model size.

> **Takeaway:** Similar to training efficiency, Textbook Consistency also scales with parameters: the gain increases with model size.

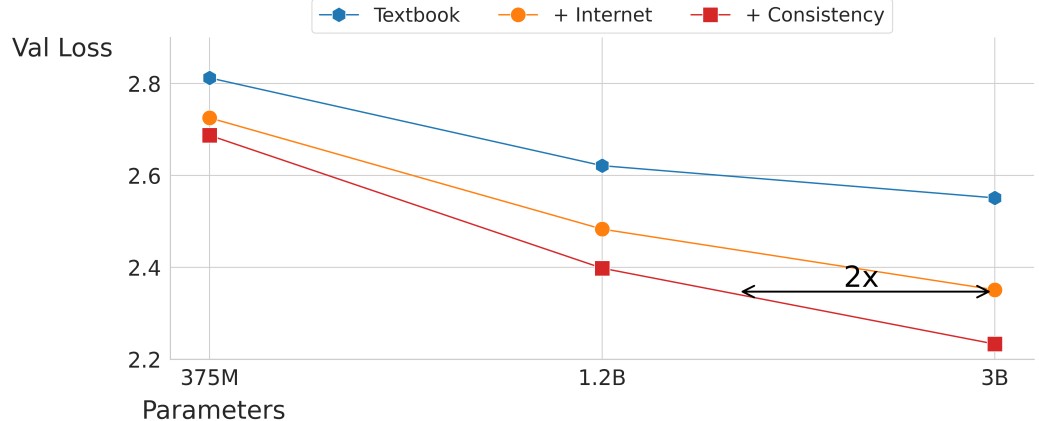

**Figure 3** **Validation loss as a function of parameters**. Parameters (375M, 1.2B, 3B) are used for models trained on textbook data (blue), textbook + internet data (orange), and textbook + internet data with consistency training (red). The inclusion of the Textbook Consistency approach significantly reduces validation loss, showing a notable 2x improvement in models with 3B parameters.

### 3.1.3 EVALUATE DOWNSTREAM TASKS

Table 1 shows the evaluation results on downstream tasks, comparing models trained on different datasets: textbook-only data, textbook plus internet data, and Textbook Consistency approach. The evaluation is based on the Eleuther AI Eval Harness (Gao et al., 2024) for standardized comparison. Table 1 reports validation loss on the textbook holdout set, along with accuracies on downstream tasks such as LAMBADA (Paperno et al., 2016), HellaSwag (Zellers et al., 2019), and NaturalQuestions (Kwiatkowski et al., 2019). The model trained with only textbook data shows the highest validation loss on the textbook holdout (2.538) and lower accuracies on the downstream tasks. Adding internet data reduces the validation loss (2.351) and improves performance across downstream tasks. However, the best results are achieved by incorporating textbook consistency, where the validation loss is further reduced to 2.233, and the model achieves the highest accuracies across all downstream tasks. These results demonstrate that the textbook consistency approach not only improves the model's generalization on the holdout dataset but also significantly boosts performance in diverse downstream tasks.

**Table 1** **Evaluation on downstream tasks.** Textbook Consistency achieves lower validation loss and higher downstream accuracy than baselines.

|  | Val Loss ($\downarrow$) | LAMBADA ($\uparrow$) | HellaSwag ($\uparrow$) | NaturalQuestions ($\uparrow$) |
|---|---|---|---|---|
| Textbook | 2.538 | 9.9 | 7.3 | 11.4 |
| + Internet | 2.351 | 11.5 | 9.8 | 13.4 |
| + Textbook Consistency | **2.233** | **12.6** | **11.5** | **14.8** |

**Takeaway:** The Textbook Consistency model, combining textbook and internet data with textbook consistency, outperforms others with the lowest validation loss and highest accuracies on downstream tasks like LAMBADA, HellaSwag, and NaturalQuestions.

### 3.1.4 EVALUATING TEXTBOOK CONSISTENCY ON THE PILE DATASET

So far, the experiments are based on the FineWeb (Penedo et al., 2024) dataset. We experimented with applying Textbook Consistency to a different internet dataset The Pile (Gao et al., 2020) to check its effectiveness. Figure 4 compares the validation loss across various domains from The Pile dataset with and without the use of textbook consistency. The orange bars represent models trained without consistency, while the red bars represent models that incorporate consistency. Across all domains, the use of textbook consistency leads to a reduction in validation loss. For example, in

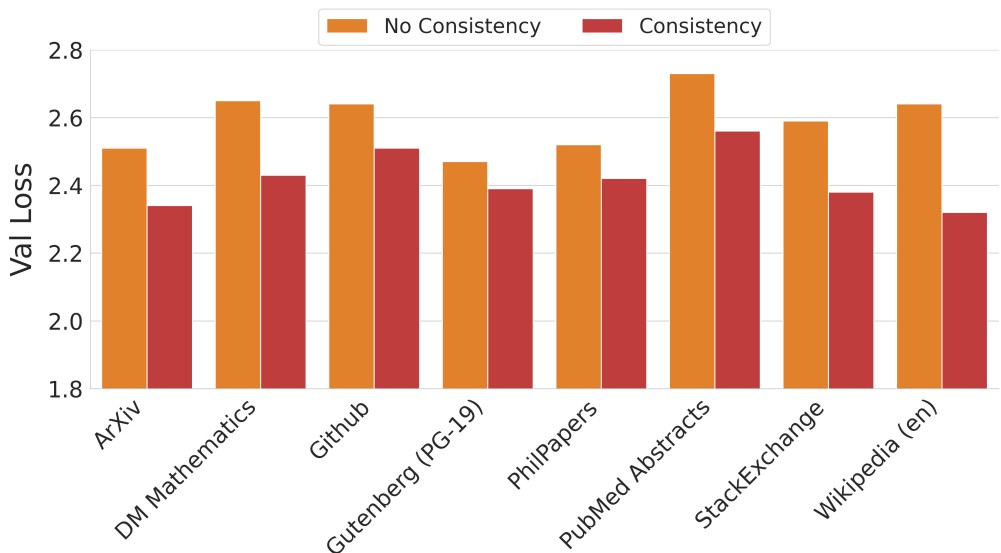

**Figure 4  Validation loss on different domains from The Pile dataset**. Textbook Consistency achieves lower validation loss than baseline.

domains such as ArXiv and DM Mathematics, the improvement is quite significant, with the red bars showing noticeably lower validation loss compared to the no-consistency baseline. Other domains, such as GitHub, Gutenberg, and StackExchange, also exhibit substantial reductions in loss when using consistency. Even in more specialized domains like PubMed Abstracts and PhilPapers, the consistency approach consistently outperforms the baseline. This demonstrates that the adaptive consistency method is effective in reducing the model's validation loss across a diverse range of domains, suggesting that it helps improve the model's generalization and capability to handle varied content more efficiently.

> **Takeaway:** Applying Textbook Consistency to models trained on The Pile dataset consistently reduces validation loss across various domains. This improvement is particularly notable in technical and specialized domains like ArXiv, Mathematics, and PubMed Abstracts, indicating that the method enhances the model's ability to generalize across diverse content areas.

### 3.1.5  EVOLUTION OF CONSISTENCY

Figure 5 illustrates the evolution of adaptive textbook consistency values during training, showing how the model adjusts the weight of training data based on its consistency with textbooks. The x-axis represents the consistency values, ranging from negative (indicating low consistency) to positive (indicating high consistency), while the y-axis tracks the evolution over time. The intensity of the color indicates the density or frequency of these values at different stages of training.

The fact that consistency is predominantly positive indicates that the FineWeb data used in the training is of high quality and generally aligns well with the textbook material. The model is effectively learning to weight more consistent data heavily, leading to improved generalization and performance on the textbook holdout and other downstream tasks. This adaptive weighting of the training data allows for more efficient and targeted learning, leveraging the strengths of both textbooks and the internet data.

### 3.2  ABLATION STUDY

Table 2 presents the results of an ablation study, comparing variations of the Textbook Consistency method to evaluate its impact on validation loss and how each variant performs against a baseline.

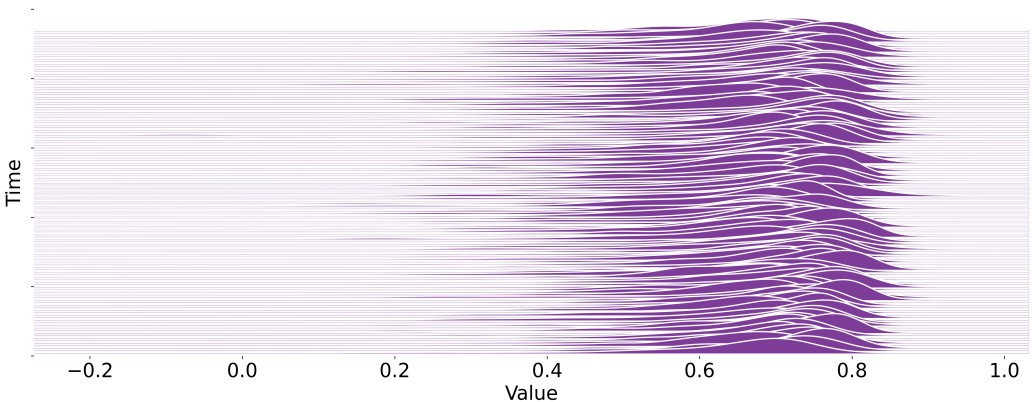

**Figure 5** **Evolution of adaptive textbooks consistency during training**. Consistency is mostly positive since FineWeb is a high quality dataset. Textbook Consistency adaptively weight training data acccording their consistency with textbooks. The training progress is from bottom to top.

**Table 2** **Ablation study**. Comparing variations of Textbook Consistency on hold out evaluation.

|  | Adaptive | Learning Rate | Target | Val Loss (↓) | vs Baseline (↓) |
|---|---|---|---|---|---|
| Default | Yes | | Textbook | **2.233** | **-0.118** |
| (A) | No (Filtering 0.6 - 0.8) No (Filtering 0.6 - 1) No (Filtering 0.4 - 0.8) | | | 2.358 2.358 2.362 | 0.007 0.007 0.011 |
| (B) | | 1.5x 0.8x 0.4x | | 2.386 2.393 2.409 | 0.035 0.042 0.058 |
| (C) | | | exclude C4 C4 | 2.268 2.354 | -0.083 0.003 |

**Data Filtering**. In configuration (A), filtering techniques are introduced, restricting the consistency target range to various thresholds (0.6-0.8, 0.6-1, and 0.4-0.8), and these variations consistently result in a higher validation loss (2.358 - 2.362), with minimal to no improvement over the baseline. This indicates that filtering consistency targets to these specific ranges limits the model's performance, possibly because it excludes some useful training data.

**Learning Rate**. Configuration (B) explores the effects of varying the learning rate (1.5x, 0.8x, 0.4x), and all variants result in higher validation losses (2.386 - 2.409) compared to the default setting. These results suggest that the default learning rate is optimal for this consistency approach, and adjusting the learning rate, either up or down, degrades performance. This shows the effectiveness of Textbook Consistency is not because adaptive weight samples may indirectly reduce learning rate.

**Textbook Source**. In configuration (C), the model is tested without C4 data and with only C4 data. The validation loss is lower when the model excludes C4 (2.268) compared to when it relies solely on C4 (2.354), but both configurations show a smaller improvement over the baseline (-0.083) or underperform baseline (0.003). This indicates that while C4 data can be useful, combining it with other sources in the default method yields the best results.

> **Takeaway:** Ablation studies show that adaptive weighting is crucial, and using textbooks is important to the method's success. Replacing adaptive weighting with filtered weights or using a reduced learning rate negatively impacts model performance.

### 3.3 APPLICATION TO SIMULATED ROBOTICS

Table 3 presents an application of the Textbook Consistency method to simulated robotics tasks using the ExoRL dataset (Yarats et al., 2022). ExoRL includes a large set of diverse yet low-

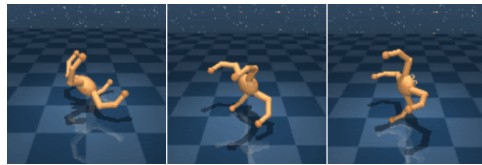

**Figure 6   ExoRL dataset includes a large set of diverse but low reward behaviors.** Textbook Consistency trains on them by checking consistency with a small set of demonstrations.

**Table 3   Rewards achieved on ExoRL.** Textbook Consistency outperforms learning from ExoRL only and learning from demonstration only.

| ExoRL | BC-10% ExoRL | BC Demo | BC Demo + BC-10% ExoRL | Textbook Consistency on ExoRL + Demo |
|---|---|---|---|---|
| Walker Stand | 52.91 | 258.52 | 127.54 | 312.34 |
| Walker Run | 34.81 | 287.44 | 108.85 | 309.85 |
| Walker Walk | 13.53 | 234.34 | 94.57 | 267.45 |
| Cheetah Run | 34.66 | 278.65 | 187.55 | 323.95 |
| Jaco Reach | 23.95 | 253.50 | 201.87 | 301.87 |
| Cartpole Swingup | 56.82 | 217.37 | 198.56 | 257.67 |
| **Total Average** | 36.11 | 254.97 | 153.16 | **295.52** |

reward behaviors, making it challenging for agents to learn optimal actions (Laskin et al., 2021). Examples of randomly sampled trajectories from the dataset are shown in Figure 6. To address this challenge, Textbook Consistency trains the model by evaluating the consistency between the low-reward behaviors from ExoRL and a small set of demonstration data we collected from the corresponding RL environments, effectively combining both sources to guide learning. This hybrid approach allows the model to leverage the strengths of both the demonstration data (high-quality examples) and the broader ExoRL dataset (diverse but noisy behaviors).

We compare behavior cloning (BC) on demonstrations with BC applied to the top 10% of high-return trajectories from diverse ExoRL (BC-10%). Both are widely used and effective approaches. Table 3 shows returns achieved across various simulated robotics tasks using different methods. The results demonstrate that Textbook Consistency outperforms all other methods on average and across most tasks. ExoRL and BC-10% ExoRL yields relatively low rewards due to ExoRL is diverse and low-reward, BC Demo outperforms BC-10% on ExoRL, combining demonstrations with ExoRL (BC Demo + BC-10% ExoRL) achieves higher performance. Textbook Consistency, which adaptively balances consistency between ExoRL and demonstration data, achieves the highest rewards overall.

> **Takeaway:** Textbook Consistency effectively combines demonstration data with diverse but low-reward behaviors from the ExoRL dataset, outperforming baselines in simulated robotics tasks.

## 4   RELATED WORK

**Data Filtering**. Data filtering has been a high-impact and active area of research for language model training (Brown et al., 2020). In addition to basics such as duplicate removal, methods include filtering data based on similarity to Wikipedia (Gururangan et al., 2022; Wenzek et al., 2019; Touvron et al., 2023a), heuristic-based (e.g., language and item count filtering), perplexity filtering, and hand curation (Penedo et al., 2023; Abbas et al., 2023; Li et al., 2024; Penedo et al., 2024). Further methods propose to filter pretraining data so that the resulting LLM will achieve higher scores on given benchmarks. This is done by selecting training data that is similar to data from a given benchmark, such as based on n-gram overlap (Xie et al., 2023), embedding similarity (Everaert and Potts, 2023), or loss-performance correlation coefficients from existing pretrained models (Thrush et al., 2024); or less scalable approaches that involve training proxy LLMs using various data mixtures (Touvron et al., 2023a; Ilyas et al., 2022; Xie et al., 2023; Engstrom et al., 2024; Liu et al., 2024). Prior research conducted extensive comparison of pretraining data selection techniques (Li et al., 2024) and found

that many of these techniques have yet to show significant improvements. The current state-of-the-art across many tasks remains fairly basic: typically, a fixed fastText (Bojanowski et al., 2017) or BERT (Devlin et al., 2018) classifier, applied after comprehensive deduplication and filtering. Another line of research focus on curriculum-based online data selection of challenging samples (Jiang et al., 2019; Loshchilov and Hutter, 2015; Katharopoulos and Fleuret, 2018) requires proxy models to determine difficulty. This approach is computationally expensive, limiting its scalability. Our work is motivated by whether we can efficiently adjust the importance of internet-sourced examples based on their consistency with high-quality textbooks. We provide compelling evidence supporting this hypothesis – demonstrating that a simple adaptive weighting method, based on textbook-internet consistency, can significantly improve training efficiency and model performance.

**Data Mixing**. Training datasets consisting of data from different domains or sources (for example, web text, code, and Wikipedia) raise an important challenge for the data curation process: determining the percentage of data that should come from each source, referred to as data mixing. These data mixing methods include using heuristics (such as human judgment) (Gao et al., 2020; Touvron et al., 2023b), or using a set of predefined configurations (Soboleva et al., 2023), or empirically determining the best domain weights according to some downstream evaluation (Du et al., 2022). Other research has explored more principled approaches (Albalak et al., 2024b; 2023; Xie et al., 2023; Thudi and Maddison, 2024) using theories such as multi-armed bandits and distributionally robust optimization. In addition, clustering-based rebalancing methods for data sampling (Shao et al., 2024) have been proposed, outperforming both uniform and other cluster-based sampling methods. Clustering-based method dynamically adjusts data sampling to rebalance data, while our method dynamically adjusts weights to be consistent with textbooks. Further methods have been proposed, including the use of learning-based strategies that optimize domain proportions through iterative training of both reference and proxy models (Fan et al., 2023), skills-based selections (Chen et al., 2024), dynamically updates the composition of sampled data based on varying losses across different domains (Xia et al., 2023), and simultaneously models the behaviors of data quantity and mixing weights using proxy models (Ge et al., 2024). These methods are primarily focused on mixing and balancing data sources. Our method takes an orthogonal approach: Textbook Consistency focuses on training time adaptive weighting using textbooks from the target domain.

## 5 DISCUSSION AND CONCLUSION

In this work, we present Textbook Consistency, a novel approach to enhancing the training efficiency of large language models by leveraging textbooks as guiding signals to adaptively weight internet-scale data. Our method dynamically adjusts the importance of data samples based on their cosine similarity to textbook content within a latent space. Empirical evaluations demonstrate that Textbook Consistency can substantially reduce training cost and improve training efficiency by more than two times, while consistently improving model performance across a wide range of benchmarks as well as non-language tasks. Our experimental results indicate that Textbook Consistency is a computationally cost-free technique that improve pretraining efficiency twofold on state-of-the-art, curated datasets. This efficiency enables further advancing and scaling both model and data sizes, and as our experiments have shown, the advantage of Textbook Consistency becomes more significant as the scale increases. Textbook Consistency offers several key advantages over conventional data curation techniques: it is straightforward to implement, introduces no additional computational overhead, and can be straightforwardly integrated with existing data filtering methods.

**Limitations**. Although our Textbook Consistency is effective, it has several limitations. Currently, it uses embedding models to measure the consistency between textbooks and internet sources, which does not take advantage of large language models, or the model being trained itself. However, this limitation could be addressed by incorporating advanced models directly. Our study also has limitations in scale. We experimented with models ranging from 375M to 3B parameters, trained on datasets from 200M to 80B tokens. While our token-to-parameter ratios (up to 30-60) exceed Chinchilla's optimal ratio 20, our largest models are still significantly smaller than contemporary LLMs, which often surpass 100B parameters. This scale disparity may limit the direct applicability of our findings to these larger models. Extrapolating our results to estimate the performance and behaviors of much larger LLMs should be done cautiously, as different scaling laws may apply beyond the ranges we explored.

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

## A  FURTHER EXPERIMENT DETAILS

We employ a Llama-like architecture for models of 375M, 1.2B, and 3B parameters. The configurations for these models are provided in Table 4.

**Table 4  Model configuration**. Comparison of model architectures across different configurations (375M, 1.2B, 3B), showing key attributes such as hidden size, intermediate size, and attention heads.

| Config | Llama-like 375M | Llama-like 1.2B | Llama-like 3B |
|---|---|---|---|
| Hidden Size | 1536 | 2048 | 3200 |
| Intermediate Size | 4096 | 5504 | 8640 |
| Hidden Layers | 12 | 24 | 26 |
| Attention Heads | 16 | 16 | 32 |
| Key/Value Heads | 16 | 16 | 32 |
| RMS Norm Eps | 1e-6 | 1e-6 | 1e-6 |

