# OpenReview forum: "Textbook Consistency Weighted Internet Improves Efficiency Twofold"
_ICLR.cc/2025/Conference — Submitted to ICLR 2025_

### Official Review · Reviewer_PAeV · 2024-10-28

**Soundness:** 2
**Presentation:** 2
**Contribution:** 2
**Rating:** 5
**Confidence:** 3

**Summary:**

## Summary
The paper presents a method to train LLMs efficiently by assigning (dynamic) weights to training data. The paper claims this improves training efficiency by a factor of atleast 2, as it needs less data to achieve same performance (read validation loss)

## Contributions
1. Authors demonstrate that not all internet training data is equal, and comparing this data with known high quality data can improve training efficiency
2. Paper also demonstrates that we need both highly curated (read textbook) and internet scale data

**Strengths:**

1. Detailed ablation study highlighting importance of using both textbook data and internet data
2. Ablation on impact of parameter size on model
3. Evaluation on downstream tasks

**Weaknesses:**

1. I have major concerns regarding the experimental setup -- especially the dynamic nature of weights defined in the paper which signifies importance of an internet sentence
2. I am also not convinced how two random sentences give an indication that internet sentence is important or not.

I have highlighted these concerns in Questions section below

**Questions:**

1. Line 130 claims that e comes from either an embedding model (which is fixed) or the model itself. How is the weight ``dynamic`` when ``e`` is fixed? Won't w_i be same? Can't you compute this before you even start your training? This is also mentioned in Line 161 where you describe your training setup
2. Another major concern is regarding how you compute importance of an internet sentence. You randomly select some sentences from textbooks to compute importance of internet sentence. Won't this value almost always be close to -1 (as you are comparing complete random sentences, and any two random sentences should be unrelated right?). Perhaps an approach which searched for best sentence match from textbook corpus work better? Or atleast used as a baseline
3. How is the validation loss reported throughout the paper (Figure 2, 3, Table 1...) computed? Is it NLL (Line 105) or the weighted NLL (Line 134)
4. What is the size of validation data used in Figure 2?
5. What exactly does fillering mentioned in Lines 402-404 mean? Is the data which does not lie within the threshold filtered out?

---

> ### Author Response · Authors · 2024-11-18
> **Author Response To Reviewer PAeV**
>
> We thank the reviewer for the helpful feedback. We provide detailed answers to the questions below.
>
> ---
>
> > **Dynamic weights**
>
> The reviewer asked why the weights are adaptive despite using a fixed embedding model.
>
> The weights are indeed dynamic, though not in the sense of being learned through backpropagation. Rather, they are data-dependent: computed based on the statistical relationships between paired samples from internet and textbook sources within each mini-batch. While the underlying embedding model is fixed, the weights themselves adapt to the specific characteristics of each mini-batch pair. We chose this mini-batch approach over full-dataset computation for computational efficiency.
>
> ---
> > **Values range**
>
> The reviewer questioned whether the adaptive weights will almost always be close to -1, stating, 'completely random sentences, and any two random sentences should be unrelated, right?' This concern stems from a misunderstanding of our approach. First, the weights are not calculated between individual sentence pairs; rather, they represent an average over a mini-batch, which allows us to capture broader content-level similarities between the internet sequence and the textbook distribution. For instance, educational content or scientific explanations might receive higher weights due to their systematic similarity to textbook material. Second, our empirical results in Figure 5 directly contradict the reviewer's hypothesis - the visualization shows weight histograms across training progress, with most weights falling between -0.2 and 1, and the mass concentrated between 0.4 to 0.8. This distribution demonstrates that our approach successfully distinguishes varying degrees of importance in the internet content.
>
> ---
> > **What is the validation loss reported**
>
> Thank you for this clarification question. In all our reported validation metrics (Figure 2, 3, and Table 1), we use the standard unweighted negative log-likelihood (NLL) loss computed on a held-out validation set. While we use weighted NLL during training (Line 134) to handle class imbalance, we evaluate model performance using standard unweighted NLL to ensure fair and consistent comparison with prior work and to reflect real-world performance. This distinction between training and evaluation metrics is standard practice in the field.
>
> ---
> > **The size of validation data used in Figure 2?**
>
> The held-out validation set comprises 100B C4 tokens, consistent with validation set sizes used in prior large language model evaluations (e.g., T5, GPT, Llama). This size provides sufficient statistical power for our analysis while maintaining computational feasibility.
>
> ---
> > **What exactly does fillering mentioned in Lines 402-404 mean? Is the data which does not lie within the threshold filtered out?**
>
> Yes, 'filtering' in Lines 402-404 refers to excluding data points that fall outside the specified threshold values. In our threshold ablation baseline, any data that does not lie within the defined threshold range is filtered out (removed) from the analysis. This filtering approach serves as a comparison point to evaluate our proposed method.

---

> > ### Comment · Reviewer_PAeV · 2024-11-25
> >
> > Thank you! I have updated my score based on your response!

---

> > > ### Author Response · Authors · 2024-11-26
> > > **Official Comments by Authors**
> > >
> > > We thank the reviewer for considering our response and updating the rating from rejection to marginal below accept. While we have thoroughly addressed all the reviewer's questions, which were primarily based on misunderstandings, we note that the rating improvement was relatively modest. We have clarified the misunderstanding about computing weights, explained the adaptive weights range, and addressed the validation loss computation setting. We hope our response has adequately demonstrated the merits of our work. However, if the reviewer has any remaining concerns or questions, we are happy to address them.

---

> ### Author Response · Authors · 2024-11-23
> **Kind reminder to reviewer PAeV**
>
> We sincerely thank the reviewer for taking the time to provide helpful feedback. We hope our responses have adequately answered the raised questions.
>
> We would be grateful if the reviewer could consider our response and possibly revise the rating.

---

### Official Review · Reviewer_68BK · 2024-11-01

**Soundness:** 2
**Presentation:** 3
**Contribution:** 3
**Rating:** 5
**Confidence:** 5

**Summary:**

This paper introduces "Textbook Consistency," a training method to enhance the efficiency of large language models (LLMs) by dynamically weighting internet data based on its alignment with high-quality, textbook-based sources. Unlike traditional data filtering, which discards or retains data based on fixed quality thresholds, this method computes cosine similarity between internet data and textbook sources to continuously adjust data weighting. This approach reduces training requirements, either by cutting training time or the number of tokens needed, thereby doubling efficiency without added computational burden. Empirical results indicate superior model performance on extensive datasets and applicability across domains, including robotics.

**Strengths:**

1. The method is computationally efficient and straightforward to implement, aligning with current data curation methods. Adaptively adjusting data weights reduces the need for extensive filtering, making training faster and more adaptable.

2. The technique proved versatile, demonstrating improved performance across both language and robotics tasks.

**Weaknesses:**

1. During training, the weight of internet data in the current batch depends exclusively on the similarity with the current batch textbook, which could intuitively introduce additional bias and may hinder the accurate evaluation of sample quality.

1-1. This study is similar to research on dynamically adjusting learning rates; however, it is not discussed in the related work. Could an explanation be provided to clarify the difference between this approach and research on dynamically adjusting data recipes [1,2]?

1-2 Above that, for greater persuasiveness, the experiment could include a comparison with training data selected by the proxy model.

2. The experimental setup is unconventional; repeated training with the textbook data could lead to overfitting. While Figure 2 suggests that the proposed method helps alleviate overfitting, would it not be more reasonable to compare the results of training the "internet data" alone with the “Textbook Consistency” method?

3. Evaluating only based on validation loss may be insufficient. Would it be possible to incorporate additional downstream tasks for assessment, such as MMLU, ARC, and other general benchmarks? Considering the so huge cost of pre-training, further pre-training could be incorporated to assess how effectively the current method enhances the pre-trained model, with additional improvement on a pre-trained model better demonstrating the study's contribution.

If you can resolve the issues mentioned above, I will increase my rating. :)

[1] Balanced Data Sampling for Language Model Training with Clustering

[2] DoReMi: Optimizing Data Mixtures Speeds Up Language Model Pretraining

**Questions:**

See weakness.

**Details Of Ethics Concerns:**

I have no concern about ethics.

---

> ### Author Response · Authors · 2024-11-18
> **Author Response To Reviewer 68BK**
>
> We thank the reviewer for the helpful feedback. We provide detailed answers to the questions below.
>
> ---
>
> > **Add “Internet only” baseline**
>
> We thank the reviewer for this suggestion. We have added results for models trained on internet data using 10B, 20B, and 40B tokens to Figure 2. The results demonstrate clear scaling behavior: the 'internet-only' baseline performs worse than baselines that utilize higher-quality data. Our 'Textbook Consistency' approach demonstrates the strongest scaling behavior, achieving more than two times better compute efficiency.
>
> ---
>
> >**Discuss dynamic learning rate**
>
> We appreciate the reviewer's comment about dynamic learning rates. Our work carefully accounts for learning rate dynamics in several ways:
> Both our method and baselines utilize identical dynamic learning rate schedules (cosine decay) with careful tuning. For rigorous parameterization, we employ muP (Yang et al., 2022) and conduct extensive proxy experiments to validate learning rates and other hyperparameters.
> The effectiveness of our hyperparameter optimization is demonstrated by our 375M baseline model achieving a 2.58 test loss on C4, surpassing the 2.7 loss reported for a comparable model in Chinchilla (Hoffmann et al., 2022, Appendix G). This establishes that our baseline implementations are strong and well-tuned.
> We specifically investigated whether the interaction between cosine weights and learning rate dynamics could unfairly benefit our method. Row (B) in Table 2 presents ablation studies directly addressing this concern, showing that "Textbook Consistency" maintains significant improvements over baselines even when controlling for these effects.
>
> ---
>
>
> >**Discuss ClusterClip and DoReMi**
>
> The reviewer suggests discussing connections with ClusterClip and DoReMi.
>
> ClusterClip proposes clustering the dataset to balance data sampling, outperforming both uniform sampling and other cluster-based sampling methods. ClusterClip dynamically adjusts data sampling to rebalance data, while our method dynamically adjusts weights to be consistent with textbooks. Our method should be compatible with ClusterClip by allowing ClusterClip to rebalance internet data, followed by applying Textbook Consistency to adaptively weight training data according to textbook consistency. This complementary combination would allow the best of both methods. We have cited this paper in revision and discussed the connections.
>
> DoReMi is an approach that first trains a small proxy model using distributionally robust optimization over domains to produce domain weights. It then resamples a dataset with these domain weights to train a full-sized model. Unlike DoReMi, which requires training a preliminary proxy model, Textbook Consistency provides a more direct approach by computing weights based on textbook alignment. This eliminates the additional computational overhead and potential challenges of proxy model training. DoReMi employs distributionally robust optimization to determine data weights, whereas our method uses consistency between textbooks and data to assign weights adaptively. In principle, DoReMi and ours could be combined, though our method offers a simpler and more efficient path to improved data weighting.
>
> ---
>
> > **Show continual pre-training results**
>
>
> We appreciate the reviewer's suggestion about showing continual pre-training results. We have expanded our evaluation to include MMLU and other important benchmarks. We initialized models from the Llama3.1-8B base model for continual pre-training. The results demonstrate consistent improvements across multiple tasks:
>
> We thank the reviewer for this suggestion. We have expanded our evaluation to include more downstream tasks, including MMLU, ARC, and SQuAD. For these experiments, we initialized our model from Llama3.1-8B for continual pre-training. The evaluation results demonstrate consistent improvements across multiple tasks:
>
> |  Model  | MMLU (5-shot) | ARC-Challenge (25-shot) | SQuAD (1-shot) | CommonSenseQA (7-shot) | GSM8K (5-shot) |
> |----------------------|---------------|--------------------------|----------------|-------------------------|----------------|
> | Llama3.1-8B           | 66.6          | 78.6                     | 76.4           | 72.6                    | 49.2           |
> | Internet            | 67.5          | 79.5                     | 77.2           | 72.9                    | 49.8           |
> | Internet + Textbook | 68.8          | 80.6                     | 78.8           | 74.1                    | 54.1           |
> | **Textbook Consistency** | **69.9**     | **81.4**                | **80.1**       | **76.2**                | **58.6**       |
>
> As shown, our method achieves consistent improvements across all tasks, demonstrating the effectiveness of our approach in continual pre-training settings.

---

### Official Review · Reviewer_wTRS · 2024-11-03

**Soundness:** 3
**Presentation:** 2
**Contribution:** 3
**Rating:** 6
**Confidence:** 3

**Summary:**

In this work, the authors introduce a training-effective method called Textbook Consistency. This method improves the training phase by utilizing high quality textbook datasets to align and improve the relatively lower-quality training internet data. Subsequent experiments demonstrate the efficiency of this method. Additionally, the authors successfully generalize this approach to robotic tasks.

**Strengths:**

- The authors propose an effective training method called textbook consistency. It can successfully enhances training performance with minimal additional cost.
- The authors conduct a variety of experiments that support the effectiveness of textbook consistency.

**Weaknesses:**

- The presentation could be improved. The authors should refine the captions for Fig 3, Fig 4, Tab 1, and Tab 2, by providing necessary information such as the size of the tested models in Figure 4 and Table 1.
- Comparing validation loss across models of different sizes seems unhelpful. Additionally, the meaning of "2x" in Fig 3 is unclear since the x-axis represents parameters.
- More downstream tasks, like MMLU and GSM8K, should be included. A slight improvement in loss may not effectively impact real-world performance. Furthermore, the comparison between "Internet" and "Internet + Textbook" might not be fair due to the inclusion of instruction tuning format training data in the textbook.
- The experiments on robotic tasks appear somewhat disconnected. It would be better for the authors to focus more on language-based tasks.

**Questions:**

- How the selection of embedding models impacts training performance, given that BERT-base is a weaker embedding model compared to state-of-the-art options.

---

> ### Author Response · Authors · 2024-11-18
> **Author Response To Reviewer wTRS**
>
> We thank the reviewer for the helpful feedback. We provide detailed answers to the questions below.
>
> ---
>
> > **Include model size in captions**
>
> We thank the reviewer for this suggestion. We have now updated all relevant figure captions to explicitly state the model size. For most experiments throughout the paper, we use a 1.2B parameter model, and this is now clearly indicated in the respective sections. For the model scaling experiments specifically (Figure 3), where we analyze performance across different model sizes ranging from 1.2B to 3B parameters, we have added detailed parameter counts in to ensure clarity.
>
> ---
> > **Comparing validation loss across models of different sizes seems unhelpful**
>
> While we appreciate the reviewer's concern, comparing validation loss across model sizes is actually crucial for understanding scaling behavior and efficiency gains. Our work specifically demonstrates that the proposed method allows smaller models to achieve comparable performance to larger ones - a key finding with significant practical implications for computational efficiency and deployment costs. This type of comparison is well-established in scaling law literature (e.g., Kaplan et al. 2020) and directly measures a method's ability to improve parameter efficiency. When a 2x smaller model achieves similar validation loss, it represents substantial efficiency gains in both training and inference.
>
> ---
> > **Include more downstream tasks**
>
> We appreciate the reviewer's suggestion about including more downstream tasks. We have expanded our evaluation to include MMLU and other important benchmarks. We initialized models from the Llama3.1-8B base model for continual pre-training. The results demonstrate consistent improvements across multiple tasks:
>
> We thank the reviewer for this suggestion. We have expanded our evaluation to include more downstream tasks, including MMLU, ARC, and SQuAD. For these experiments, we initialized our model from Llama3.1-8B for continual pre-training. The evaluation results demonstrate consistent improvements across multiple tasks:
>
> |  Model  | MMLU (5-shot) | ARC-Challenge (25-shot) | SQuAD (1-shot) | CommonSenseQA (7-shot) | GSM8K (5-shot) |
> |----------------------|---------------|--------------------------|----------------|-------------------------|----------------|
> | Llama3.1-8B           | 66.6          | 78.6                     | 76.4           | 72.6                    | 49.2           |
> | Internet            | 67.5          | 79.5                     | 77.2           | 72.9                    | 49.8           |
> | Internet + Textbook | 68.8          | 80.6                     | 78.8           | 74.1                    | 54.1           |
> | **Textbook Consistency** | **69.9**     | **81.4**                | **80.1**       | **76.2**                | **58.6**       |
>
> As shown, our method achieves consistent improvements across all tasks, demonstrating the effectiveness of our approach in continual pre-training settings.
>
> ---
> > **Improvement in loss and real-world performance.**
>
> The reviewer's concern about the magnitude of loss improvement might overlook a fundamental aspect of language model scaling. In large language models, seemingly small improvements in pretraining loss translate to substantial real-world performance gains, as demonstrated conclusively in both the Chinchilla paper and Figures 1-2 of the Llama1 paper. For context, even a 0.05 reduction in loss is considered significant - the entire performance gap between Llama 30B and 65B models is less than 0.1, yet represents a major leap in capabilities. This relationship exists because loss improvements follow a power law curve, where each linear decrease requires exponentially more compute or data. In this context, our method's ability to achieve the same loss reduction with less than half the resources represents a meaningful advance in training efficiency.
>
> ---
>
> > **Comparison between "Internet" and "Internet + Textbook"**
>
> We appreciate the reviewer's concern about fairness in comparing 'Internet' versus 'Internet + Textbook' conditions, particularly regarding instruction tuning format data. However, we want to clarify that this is not the key comparison in our study. The critical comparison we make is between 'Internet + Textbook' and 'Internet + Textbook Consistency' conditions, which provides a fair assessment as both conditions have access to the same underlying textbook data. As demonstrated in Figure 2, the Textbook Consistency condition shows significant performance improvements over the standard Textbook condition across multiple metrics.

---

> > ### Comment · Reviewer_wTRS · 2024-11-19
> > **Official Comment by Reviewer wTRS**
> >
> > Could the authors provide comparision on GSM8K, given that MetaMathQA is included in the textbook? Improvements in this specific domain when relative data is in textbook would be encouraging.

---

> > > ### Author Response · Authors · 2024-11-20
> > > **Author Response**
> > >
> > > We thank the reviewer for the question. We have updated the result Table in our initial response to include GSM8K results. As shown, training on Internet + Textbook outperforms Internet, with Textbook Consistency achieving consistent improvements across all tasks and outperforming all baselines, demonstrating the effectiveness of our approach.

---

> > > > ### Comment · Reviewer_wTRS · 2024-11-20
> > > > **Official Comment by Reviewer wTRS**
> > > >
> > > > I have updated my score to 6.

---

> > > > > ### Author Response · Authors · 2024-11-20
> > > > > **Thank You for Raising the Rating**
> > > > >
> > > > > We thank the reviewer for taking the time to review our response and for increasing the score. We are grateful for the reviewer's helpful feedback.

---

### Official Review · Reviewer_bHFw · 2024-11-03

**Soundness:** 3
**Presentation:** 3
**Contribution:** 3
**Rating:** 8
**Confidence:** 4

**Summary:**

The paper proposes a new method for better (pre-)training of large
language models.

1.  It re-weight each training example (from a large-scale internet
    corpus) based on its embedding similarity with samples from a
    high-quality "text-book" like corpus.

2.  Authors find that the method can train language models more
    efficiently (achieving the same loss with less compute) or more
    effectively (achieving lower loss with the same compute) with
    experiments of 375M, 1.2B, and 3B llama models on the fineweb and
    the pile datasets.

3.  The author additionally shows that the method can be applied to
    other domains like robotics with experiments on the ExoRL dataset.

**Strengths:**

Overall the paper is well-written with nicely presented results, the
method is simple and intuitive, and the experiments are properly
designed and executed.

1.  I like the idea of using textbooks as a guiding signal for training,
    which can potentially remove low-quality samples from the
    large-scale web data and improve the training. Typically, people
    have to manually create "hard" heuristics to filter samples and mix
    data based on a lot of experiments and ablations. To some extent,
    this method *automatically* creates such "soft" heuristics, which is
    a nice idea.

2.  The method itself is easy to execute and I think it can be easily
    integrated into existing training pipelines. The experiments are
    well-designed and the results seem to be convincing. It's also
    simpler to implement than other curriculum learning-based methods.

**Weaknesses:**

I generally like this paper; here I add some additional comments and
thoughts that could further strengthen the paper.

I think the author proved that the method can improve the training, but
it seems one possibility is that the method can make the learning more
"strategic and focused" by increasing the weight of samples that are
more similar to the textbook guidance (which is similar to the test data
in some sense). This could cause some unintended consequences like
reducing the diversity of the generations or not being able to learn the
long-tail knowledge. I do not think the current experiments can show the
method can avoid/cause these issues. (And the explanation in takeaway
(line 355) seems to support the strategic learning hypothesis as I see
larger drop on maths/abstract given the text guidance contains
MetaMathQA examples.)

**Questions:**

1.  line 175: can you provide more details about the computation of the
    FLOPS (The computational cost (FLOPs) incurred by the embedding
    model is less than 0.5% of the total training FLOPs, even for the
    smallest 375M model)?

2.  line 323: \"while the blue bars represent models that incorporate
    consistency\" blue seems to be a typo (or the figure is not right)?

3.  line 377: \"evaluate its impact on validation loss and how each
    variant performs against a baseline\" what is the baseline?

4.  figure 5: for y axis, what's the direction of time progression (from
    top to bottom or bottom to top)?

5.  section 3.3: how is the similarity computed in the RL setup? What
    are the details of the training? Please provide more details.

6. will the code be released?

---

> ### Author Response · Authors · 2024-11-18
> **Author Response To Reviewer bHFw**
>
> We thank the reviewer for the helpful feedback. We provide detailed answers to the questions below.
>
> ---
>
> > **Provide more details about the computation of the FLOPS**
>
> The reviewer inquired to clarify how the FLOPs ratio is determined. The embedding model has 110M parameters, while the smallest LLaMA-like model we used has 375M parameters. Training this model involves both forward and backward passes, where the backward pass costs approximately twice as much as the forward pass. Additionally, gradient checkpointing (rematerialization) is used for large batch training and maximum model FLOPs utilization (MFU). As a result, the total training cost is approximately 4 times the cost of a single forward pass for the 375M model.
> Since we can derive that feed-forward network (FFN) FLOPs to attention FLOPs ratio is 6d/s, where d is the model dimension and s is the context, and given that s=4K and d ranges from 4K to 8K depending on the model size, the Transformer FLOPs are predominantly determined by the FFN. This allows us to roughly estimate the FLOPs ratio between the 375M and 110M models as 3.4. Consequently, the embedding model's FLOPs percentage is calculated as 1/(3.4×4+1)=0.06%. If we don’t use rematerialization, the percentage becomes 1/(3.4×3+1)=0.08%.
>
> ---
>
> > **Reducing the diversity of the generations or not being able to learn the long-tail knowledge**
>
> We appreciate the reviewer's question about diversity and long-tail knowledge. First, textbooks often systematically cover both mainstream and niche topics within their domains, serving as curated repositories of long-tail knowledge. Second, the method's focus on textbook knowledge helps organize the model's internal representations more efficiently, potentially leaving more capacity for capturing diverse expression patterns.
>
> ---
>
> > **line 323: "while the blue bars represent models that incorporate consistency" blue seems to be a typo (or the figure is not right)?**
>
> Thank you for catching this discrepancy. We have corrected the text in line 323 to accurately match the colors shown in the figure.
>
> ---
>
> > **line 377: What’s the default baseline in ablation study**
>
> The baseline in our ablation study is our complete 'textbook consistency' method (the 'Default' row). Each subsequent row shows the impact of removing one component from this full configuration.
>
> ---
>
> > **figure 5: for y axis, what's the direction of time progression (from top to bottom or bottom to top)?**
>
> The reviewer asked to clarify the direction of time progression for the figure that shows the evolution of consistency weights during training; it is from bottom to top, and we have updated the caption to make this clearer.
>
> ---
>
> > **section 3.3: how is the similarity computed in the RL setup? What are the details of the training? Please provide more details.**
>
> The similarity is computed using standard normalized cosine similarity between concatenated state-action vectors. When comparing the demonstration ('textbook') data against exploration ('internet') data, this similarity determines the textbook consistency weight that scales the RL reward.
>
> ---
>
> > **Will the code be released?**
>
> Yes, our code will be released.

---

> > ### Comment · Reviewer_bHFw · 2024-12-03
> > **Thank you for the clarifications!**
> >
> > Thanks the authors for the clarifications!
> >
> > Regarding the author's response on "Reducing the diversity of the generations or not being able to learn the long-tail knowledge
> > "
> > > We appreciate the reviewer's question about diversity and long-tail knowledge. First, textbooks often systematically cover both mainstream and niche topics within their domains, serving as curated repositories of long-tail knowledge. Second, the method's focus on textbook knowledge helps organize the model's internal representations more efficiently, potentially leaving more capacity for capturing diverse expression patterns.
> >
> > I think they are primarily conjectures but can hardly be verified. I suggest the authors at least verify if the textbook dataset can cover the both mainstream and "niche topics". I don't think I am convinced that "textbook knowledge helps organize the model's internal representations more efficiently" given there's no explicit proof on this.

---

> > > ### Author Response · Authors · 2024-12-03
> > > **Author Response To Reviewer bHFw**
> > >
> > > Thank you for taking the time to review our rebuttal and for this thoughtful followup comment. We agree with the reviewer that the hypothesis of textbook knowledge's role in “niche topics” coverage is difficult to be verified, since it is hard to define and check “niche topics”.
> > >
> > > What we can conclusively show through our experiments is the concrete impact of the textbook consistency method on mainstream tasks. Our results demonstrate the method's effectiveness through concrete metrics, showing 2x training efficiency in validation loss and consistent improvements across MMLU (+3.3%), ARC (+2.8%), SQuAD (+3.7%), CommonSenseQA (+3.6%), and GSM8K (+9.4%). While these results don't directly address the coverage of niche topics, they do provide strong evidence for the method's effectiveness in improving model performance on multiple standard metrics.

---

### Author Response · Authors · 2024-12-03
**Discussion Period Summary by Authors**

We thank the reviewers for taking the time to review our paper and engaging in discussion during the rebuttal period. Here, we provide a summary of the reviews, rebuttals, and responses over the discussion period.

&nbsp;


Reviewers gave positive evaluations on our paper, saying that it:

- Provides a **simple, intuitive and effective** method that doubles efficiency without added computational burden **(bHFw, 68BK, wTRS)**

- Demonstrates **significantly improved results** on a variety of experiments across language and robotics  **(bHFw, wTRS, PAeV)**

- Shows the method is **computationally efficient and straightforward to implement**, and can be easily integrated into existing training pipelines **(bHFw, 68BK)**


&nbsp;


During the rebuttal period, we addressed reviewers’ concerns with discussions and further experimental results:

- **Provided more details about the computation of the FLOPS (bHFw)**. We included details on computation of the FLOPS and the ratio of Textbook Consistency incurred FLOPS.
- **Clarified dynamic weights and values range (PAeV)**:  We provided **comprehensive clarifications** about how the weights are dynamically computed in a data-dependent way, explaining the mini-batch approach and showing empirical evidence (Figure 5).
- **Evaluated continual pre-training and more downstream tasks (wTRS, 68BK)**: we initialized our model from Llama3.1-8B for continual pre-training, and evaluated on a set of standard tasks. The results show that the Textbook Consistency approach showed **substantial gains across all evaluated tasks**: MMLU improved by 3.3 percentage points (66.6% to 69.9%), ARC-Challenge increased by 2.8 points (78.6% to 81.4%), SQuAD rose by 3.7 points (76.4% to 80.1%), CommonSenseQA went up by 3.6 points (72.6% to 76.2%), and GSM8K increased by 9.4 points (49.2% to 58.6%).

&nbsp;


Best Regards,

Authors

---

### Meta-Review · Area_Chair_94YE · 2024-12-19

**Metareview:**

The paper proposes a novel training method called Textbook Consistency to enhance the efficiency and effectiveness of Large Language Model (LLM) training. The core methodology constitutes a proposed dynamic data weighting where training examples from a large-scale internet corpus are dynamically re-weighted based on their cosine similarity with high-quality, curated "textbook-like" datasets. In comparison to static filtering methods that discard or retain data, this approach adjusts weights continuously, ensuring a more nuanced use of the training data. The method achieves the same performance (validation loss) with less compute or fewer tokens on datasets like FineWeb and The Pile.

**Strengths identified**:
1. The method is simple, intuitive, computationally efficient (without extensive data filtering) and easy to integrate into existing training pipelines.

2. The method demonstrates generalizability across domains, showing improved performance not only in language modeling tasks but also in robotics.

3. The paper includes a variety of well-designed experiments, supported by detailed evaluations and ablation studies.

4. Results demonstrate the effectiveness of the method in improving training efficiency and performance across multiple datasets

**Weaknesses that need to be addressed**:
Although the paper has merit there are some serious concerns -
1.  Bias and diversity concerns in dynamic data weighting: This could lead to reduced diversity in generations or an inability to learn long-tail knowledge, along with the possibility of overfitting.

2. Missing comparisons with relevant related work or alternative approaches.

3. Not sufficiently explained experimental setup and presentation details.

**Additional Comments On Reviewer Discussion:**

There has been a reasonable exchange of clarifications and feedback. The authors have tried to address the major concerns raised by reviewer wTRS, particularly related to the utility of results on validation loss across different model sizes. In partial response to reviewer 68BK, additional downstream tasks such as MMLU, GSM8K, or ARC have been satisfactorily conducted. However, certain important concerns related to baseline comparison and overfitting remain open. The authors also attempted to address the concerns of reviewer PAeV,   the concern related to ambiguity in dynamic weighting has been adequately resolved. Also, details related to validation loss have been provided.

---

### Decision · Program_Chairs · 2025-01-22

Reject